# Settlement, environment, and climate change in SW Anatolia: Dynamics of regional variation and the end of Antiquity

**Matthew J. Jacobson**[1,2]*, **Jordan Pickett**[3]*, **Alison L. Gascoigne**[4], **Dominik Fleitmann**[5], **Hugh Elton**[6]

**1** Department of Archaeology, School of Archaeology, Geography and Environmental Science, University of Reading, Reading, United Kingdom, **2** Department of Archaeology, School of Humanities, University of Glasgow, Glasgow, United Kingdom, **3** Department of Classics, University of Georgia, Athens, Georgia, United States of America, **4** Archaeology, Humanities, University of Southampton, Southampton, United Kingdom, **5** Department of Environmental Sciences, University of Basel, Basel, Switzerland, **6** Ancient Greek & Roman Studies, Trent University, Peterborough, Ontario, Canada

\* matthew.jacobson@glasgow.ac.uk (MJJ); jordan.pickett@uga.edu (JP)

## Abstract

This paper develops a regional dataset of change at 381 settlements for Lycia-Pamphylia in southwest Anatolia (Turkey) from volume 8 of the *Tabula Imperii Byzantini*–a compilation of historical toponyms and archaeological evidence. This region is rich in archaeological remains and high-quality paleo-climatic and -environmental archives. Our archaeological synthesis enables direct comparison of these datasets to discuss current hypotheses of climate impacts on historical societies. A Roman Climatic Optimum, characterized by warmer and wetter conditions, facilitating Roman expansion in the 1st-2nd centuries CE cannot be supported here, as Early Byzantine settlement did not benefit from enhanced precipitation in the 4th-6th centuries CE as often supposed. However, widespread settlement decline in a period with challenging archaeological chronologies (c. 550–650 CE) was likely caused by a "perfect storm" of environmental, climatic, seismic, pathogenic and socio-economic factors, though a shift to drier conditions from c. 460 CE appears to have preceded other factors by at least a century.

## Introduction

Past climatic and environmental conditions are often considered influential for socio-cultural change in Anatolia and the broad Eastern Mediterranean-Middle East region (e.g., [1–3]. Large-scale analyses are often necessitated, but also hindered, by low availability of high-quality comparable interdisciplinary datasets proximate to one-another [4, 5] (for a strong smaller-scale example, see the Negev desert [6, 7]). This presents difficulties in analysis as both climatic and socio-economic conditions display high spatial and temporal variability [4, 8, 9]. Here, we develop a regional settlement change dataset for Lycia and Pamphylia in SW Turkey (Fig 1), adapted from volume 8 of the *Tabula Imperii Byzantini* (TIB 8: [10])—a compilation of historical toponyms and associated archaeological research [11]. The region is rich in the

**Data Availability Statement:** All relevant data are in the Supporting information files, or previously published. Paleoclimate datasets were collected from the NOAA Paleoclimate Database (https://

www.ncei.noaa.gov/access/paleo-search/). Pollen datasets were collected from the European Pollen Database (http://www.europeanpollendatabase. net).

**Funding:** This work was supported by the AHRC South, West and Wales Doctoral Training Partnership (Grant AH/L503939/1 to M. J. Jacobson). Additional support was provided by Franklin College of Arts and Sciences and the Willson Center for Humanities and Arts, both at the University of Georgia. The funders had no role in study design, data collection and analysis, decision to publish, or preparation of the manuscript. There was no additional external funding received for this study.

**Competing interests:** The authors have declared that no competing interests exist.

archaeological remains of cities, harbors, and rural settlements, in addition to high-quality paleo-climatic and -environmental datasets. Previously, the archaeological evidence was disjointed due to a lack of data-synthesis [12].

Following the production of our settlement dataset, available evidence in Lycia-Pamphylia meets a sufficient standard to test hypotheses linking climatic and socio-cultural change. Abundant paleo-environmental datasets (15 pollen records, from 9 sites) located at varied altitudes (Fig 1), detail past ecological conditions and the intensity of human agricultural activities [14–16]. Pollen in these records detail a period of human-induced land-cover change known as the Beyşehir Occupation Phase (BOP), characterized by reduced presence of forest taxa and increased presence of cultivated trees [17]. For analysis in this paper, we utilize the presence of cultivated trees, represented by OJCV (*Olea* (Olive), *Juglans* (Walnut), *Castanea* (Chestnut) and *Vitis* (Grapevine)) pollen, calculated as an average of standardized percentages for archaeological periods and overlapping 200-year time windows, as a proxy for anthropogenic influence/ intensity of agricultural activities [16, 18]. The potential agricultural productivity of land is heavily determined by location (including elevation) and climatic factors. In SW Turkey, effective moisture (soil moisture available to plants) is the primary climatic constraint to plant growth and, therefore, agricultural productivity [19, 20]. The recent publication of a highly-resolved speleothem (stalagmite) proxy record from Kocain Cave, ~38km north of Antalya, enables detailed reconstruction of regional effective moisture and precipitation amount for >3,000 years [21]. Measured ratios of magnesium with calcium (Mg/Ca) record changes in

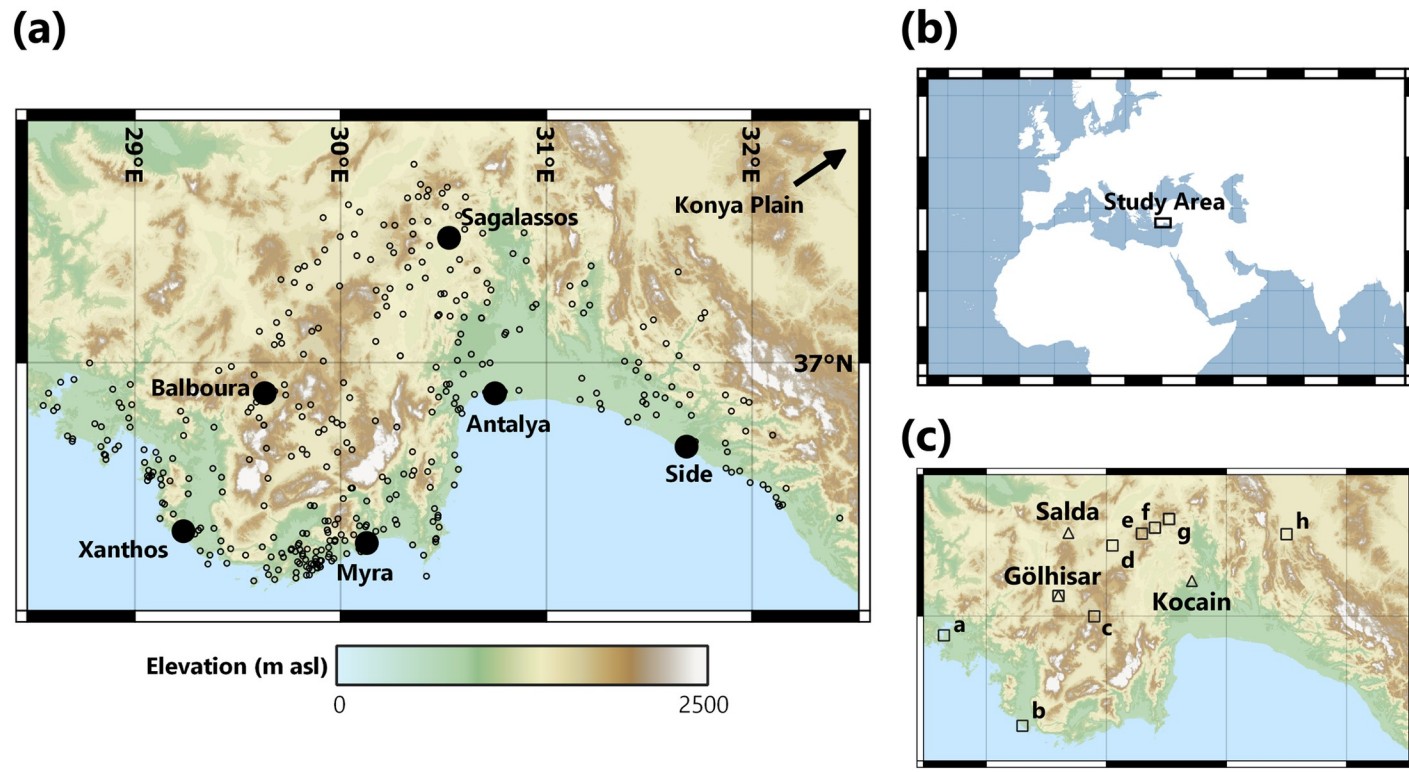

**Fig 1. Maps of the study region.** These were created in QGIS using the ASTER Global Digital Elevation Model v3 as a basemap [13]: (**a**) All TIB 8 settlements, with elevation data displayed using a topographic color ramp. Important locations are named, ancient (modern) names: Xanthos (Letoön), Balboura (Çölkayığı), Myra (Demre), Attaleia (Antalya), Side (Manavgat). (**b**) Location of study region within the Eastern Mediterranean. (**c**) Lycia-Pamphylia with paleoclimate records (triangles, named) and pollen records (squares), referenced in-text. Pollen records are: Köyceğiz (a), Ova (b), Söğüt (c), Pınarbaşı (d), Bereket Basin (e), Gravgaz Marsh (f), Ağlasun (g) and Beyşehir (h).

effective moisture at a very high resolution (>1 sample per year). Variations in oxygen isotope ratios ($\delta^{18}O$) provide information about past precipitation at decadal timescales (average of 9 years between samples). Two other paleoclimate records are in our study region and cover the period under discussion. The lower-resolution Lake Gölhisar $\delta^{18}O$ record [22], located within the Lycian Taurus mountains at 930m asl, reflects lake-water balance, a variable akin to effective moisture, on a sub-centennial timescale (average of 71 years between samples for 1,000 BCE– 1,050 CE). The multi-proxy Lake Salda record [23], located in Burdur Province at 1180m asl, extends back to ~560 CE and reflects decadal-scale changes (average of 12 years between samples for 560–1050 CE) in lake-water balance. These records detail high climatic variability during the Late Holocene, with numerous dry and wet phases identified. Our record of settlement change is compared diachronically to these high-quality paleo-climatic and -environmental datasets to examine the influence of climate change on settlement patterns.

## Study area geography

Our study region, covered by the TIB 8, pertains to the ancient regions of Lycia and Pamphylia, as well as parts of southern Pisidia, which are contained within 28˚30'-32˚30'E and 36˚00'-38˚00'N (Fig 1). The Western Taurus mountains form an orographic barrier to southerly and south-westerly airflows, separating the coastal and interior plateau regions into zones with distinct microclimates, ecological conditions, and high numbers of endemic species [24–26]. According to the Köppen-Geiger climate classification system, coastal regions are categorized as hot-summer Mediterranean (Csa) climates, with high levels of precipitation that exhibit a clear winter peak; inland regions are characterized by cold semi-arid (Bsk) and Mediterranean-influenced hot/warm-summer humid continental (Dsa/b) climates [27–29]. Colder conditions, with reduced precipitation (~33% less) characterize the regions further inland and at higher elevations [30, 31]. Fertile river valleys (e.g., the Dalaman and Köprüçay) that cut through the mountains are ideal for both rain-fed and irrigated agriculture. These run from the elevated Anatolian plateau, which is dotted with lacustrine basins and upland sheep and goat pastures called yaylas, down to indented coastlines that provide convenient natural harbors for maritime trade [32–35].

## Methods

We tabulated entries in Lycia-Pamphylia from the TIB 8 to create a settlement record spanning six established historical periods: the Bronze Age (3000–1150 BCE), Iron Age (1150–350 BCE), Hellenistic Period (350–50 BCE), Roman Period (50 BCE– 350 CE), Early Byzantine Period (350–600 CE) and Middle Byzantine Period (600–1050 CE). The TIB is a long-term project organized by the Austrian Academy of Sciences since 1966, which has coordinated historical and modern toponyms for provinces of the Roman-Byzantine Empire with an extensive compilation of primary historical sources for all periods and relevant languages, ancient inscriptions, traveler accounts from the 19th and 20th centuries, and published data from extensive archaeological surveys and excavations [11]. Our study ends at the beginning of the Turkish conquests in the 11th century CE, when the region became politically fragmented and the nature of both archaeological and textual evidence changed radically [10, 36]. The history and bibliography of each toponym was reviewed, including new publications since the TIB 8 was published in 2004, to attribute presence or absence of evidence to the six historical periods (Table 1). The type and nature of evidence from each site was also noted according to 5 categories, in order of spatial and chronological certainty (Table 2):

**Table 1. Settlement metadata by period.** "New" settlements are those with evidence in the period, but no evidence in the preceding period; "Continued" settlements are those with evidence in the period and preceding period; "Abandoned" settlements are those with no evidence in the period, but evidence in the preceding period; time-adjusted settlement numbers are calculated following the methodology of [39, 42].

| Period | Length (years) | Total Settlements | New Settlements | Continued Settlements | Abandoned Settlements | Time-Adjusted | Avg. Elevation (m asl) |
|---|---|---|---|---|---|---|---|
| Bronze Age (3000–1150 BCE) | 1850 | 25 | 25 | N/a | N/a | 3.38 | 759 |
| Iron Age (1150–350 BCE) | 800 | 114 | 100 | 14 | 11 | 35.63 | 554 |
| Hellenistic (350–50 BCE) | 300 | 205 | 104 | 101 | 13 | 170.83 | 615 |
| Roman (50 BCE– 350 CE) | 400 | 276 | 116 | 160 | 45 | 172.5 | 664 |
| Early Byzantine (350–600 CE) | 250 | 255 | 66 | 190 | 87 | 255 | 575 |
| Middle Byzantine (600–1050 CE) | 450 | 135 | 17 | 118 | 137 | 75 | 562 |

1. <u>New or restored architecture</u>—including monumental and public or domestic structures, and fortifications.

2. <u>Architectural contraction</u>—pertaining to church contractions in the Middle Byzantine period only, where communities constructed a smaller chapel within the confines of an older church complex [37].

3. <u>Material culture</u>—mainly ceramics, also tombs and graves, coins and inscriptions located within settlements.

4. <u>Textual reference</u>—settlements referenced by ancient primary sources, whether in literature, or by inscriptions and mint marks of coins found outside the named settlement (e.g., the tribute lists of the Delian League).

5. <u>Spolia</u>—older construction materials that have been reused in ancient, medieval, or modern settlements. In large quantities from ancient and medieval contexts these portend earlier settlement at a site; settlements with limited quantities of spolia in early modern contexts that may have been transported or some distance have not been included.

Settlement toponyms with archaeological or primary source evidence from at least one of the six historical periods were compiled, with their locations and elevations identified using Google Earth satellite imagery and plotted in QGIS (full dataset available in S1 Table). Compiled settlements were also ascribed measures of resolution based on how accurately they could be positioned spatially and chronologically (Table 3). No permits were required for the described study, which was exclusively desk-based and complied with all relevant regulations.

## Data critique

Significant interpretative challenges are presented when producing a settlement dataset from historical toponyms. These are analogous to those well-understood for archaeological survey

**Table 2. Settlement evidence by period.**

| Period | Architectural | Material Culture | Negative Architectural | Literary Reference | Spolia |
|---|---|---|---|---|---|
| Bronze Age (3000–1200 BCE) | 3 | 20 | | 2 | 0 |
| Iron Age (1200–300 BCE) | 63 | 30 | | 19 | 2 |
| Hellenistic (300–50 BCE) | 136 | 42 | | 24 | 3 |
| Roman (50 BCE– 350 CE) | 147 | 101 | | 11 | 17 |
| Early Byzantine (350–600 CE) | 213 | 23 | | 7 | 12 |
| Middle Byzantine (600–1050 CE) | 45 | 2 | 72 | 10 | 6 |

**Table 3. Measures of resolution.** These are visually presented in S1 Fig.

| Measure of Resolution | Explanation | Number of Settlements |
|---|---|---|
| Spatial | | |
| 1 | Site located with 1km certainty | 291 |
| 2 | Site located within 1-3km certainty | 76 |
| 3 | Site located with >3km certainty | 14 |
| Chronological | | |
| 1 | Site has been excavated or systematically surveyed and published | 56 |
| 2 | Site has been extensively surveyed by scholars; often reliant on architectural description and broad period indicators | 226 |
| 3 | Site has been summarily visited by scholars; no formal excavation or survey; or, reused architectural elements of a given period noted in medieval-modern settlement | 99 |

data ([38–41]; references below). Our dataset will reflect part of the original distribution of settlements, but understanding biases produced by chronological challenges and varied preservation is important to determine the accuracy of the pattern. There are four main issues present:

Firstly, the six periods utilized in this study vary in length from 250 to 1850 years, which will lead to overemphasis of settlement numbers during longer periods. Time-adjusted settlement numbers can help to compensate for this bias, these are calculated (following the methodology of [39, 42]) as follows:

$$Time\ adjusted\ settlement\ number = \frac{Total\ settlement\ number}{(Period\ length/Shortest\ period\ length)}$$

The shortest period length in our study is the Early Byzantine Period (250 years). In our discussion, we focus on periods with relatively short and similar durations.

Secondly, categorizing settlements into periods by relative dating (periodization/ "time-averaging") brings inherent bias. This makes settlements appear to exist for the entire period (the "synchronistic paradigm"; [43]). Thus, they are only given a *terminus ante* and *post quem* at the start and end of the archaeological period. For many settlements, particularly in rural locations, it is impossible to tell whether they were occupied throughout a particular period, for a short duration, or abandoned and reoccupied [44]. Settlements can therefore appear contemporaneous though they were not (the "contemporaneity problem"; [45, 46]). The precise definition of chronological boundaries also obscures the uncertainty associated with archaeological chronologies, which rarely correspond perfectly to one-another. For instance, the Lycian-type tombs and sarcophagi that dot the landscape remain little changed between the Late Iron Age into the Early Roman period. Different strategies and resolutions for periodization are utilized in different academic disciplines, such as by ceramicists when compared to historians.

Thirdly, destruction of evidence in later periods and time-dependent degradation will lead to an overemphasis of settlement numbers in more recent periods and alter the settlement pattern via a "preservation bias" [47]. Evidence in a particular period determines its archaeological visibility. Standing architectural remains, which constitute the bulk of archaeological evidence in our study region since the Iron Age, have an exceptional degree of preservation in Lycia-Pamphylia due to the region's relative inaccessibility and low population density. Preservation has contributed to a long history of archaeological exploration (e.g., [48]) via extensive survey and architectural description, at the expense of excavated ceramics whose publication is

more limited. Ceramics are generally robust, but different raw materials and production processes determine the level of survival, and visibility, that can change the number of sherds surviving from each period: red-slip wares from the Late Hellenistic to Late Roman periods are better represented in publication, for instance [49]. In addition to the archaeological taphonomy of settlements, biases impacting the survival of ancient site toponyms may influence our record. Whilst we include some settlements with archaeological remains that can only be identified by a modern Turkish toponym, ancient toponym survival is comparatively strong in our study region, caused by the continued presence of Greek communities until population exchanges after 1923 [50]. However, the literary and historical record does yield additional ancient toponyms that cannot be securely located on the landscape, and there are likely many smaller settlements whose toponyms did not survive, left minimal evidence, or were subsequently destroyed.

Finally, indications of settlement or population size, whose estimates are typically dependent on measurable area within city walls or counts of housing units [51], are beyond the scope of this paper. A related problem is the nature of the historical *polis*, commonly translated as "city". The polis was a rank of settlement since ~600 BCE that signaled the presence of municipal government and elite investment in monumental architecture [52]. However, such settlements did not necessarily have high-density populations and most Lycian-Pamphylian cities lack substantial domestic architecture and never reached the size, density, or population, of Ephesos, Antioch, or even Tarsos [53, 54]. In the Early Byzantine period the rank of polis was extended to more settlements, even as agrarian and artisanal-industrial activities were introduced within city walls [55] (see Discussion). In the Middle Byzantine period, many *poleis* retained their status even in cases where the archaeological evidence suggests they were small villages (e.g., Pinara or Pednelissos; see Discussion). The extent to which Lycian-Pamphylian *poleis* or "cities" may be classified as low- or high-density urban environments is therefore irresolvable in many cases [56]. In our metadata, increasing settlement density across the study region could therefore represent expansion or fragmentation, with associated population increases or decreases. Settlement types, so far as they can be reconstructed, are included in our full data (S1 Table).

## Results

Within our study area, 381 out of the total 1038 toponym entries in TIB 8 could be located and identified as settlements with sufficient survival or publication to enable identification of historical presence between the Bronze Age and the Middle Byzantine period (Table 1 and Fig 2). The remaining 658 entries were either duplicates, dated to later periods only, infrastructure not associated with settlement (e.g., bridges), natural phenomena (e.g., rivers, mountains), regional toponyms, or their location was unknown. Toponyms associated with settlements are varied, including cities, farmsteads, peri-urban harbors and temples, hilltop fortifications, island villages and caves. Settlement function, size, and status changed over time. Of 381 settlements, only eight were occupied in all six periods and 283 (75% of the total) contained evidence for 1–3 periods (S1 Fig). Elevation of settlements varied overall, ranging from 0 to 1835m asl, and by period (Fig 3). However, lower elevation settlements dominate, with 29.4% at elevations <200m asl and 70% at elevations <1,000m asl. Most settlements (291) could be located to <1km and only a very small number (14) could not be located to <3km (Table 3). Chronological resolution of settlements was weaker, with only 56 excavated or systematically surveyed and published. A total of 226 settlements were extensively surveyed but with reliance on rather broad and primarily architectural indications for chronology. The remaining 99 have weak chronological resolutions, being only summarily visited by travelers or scholars

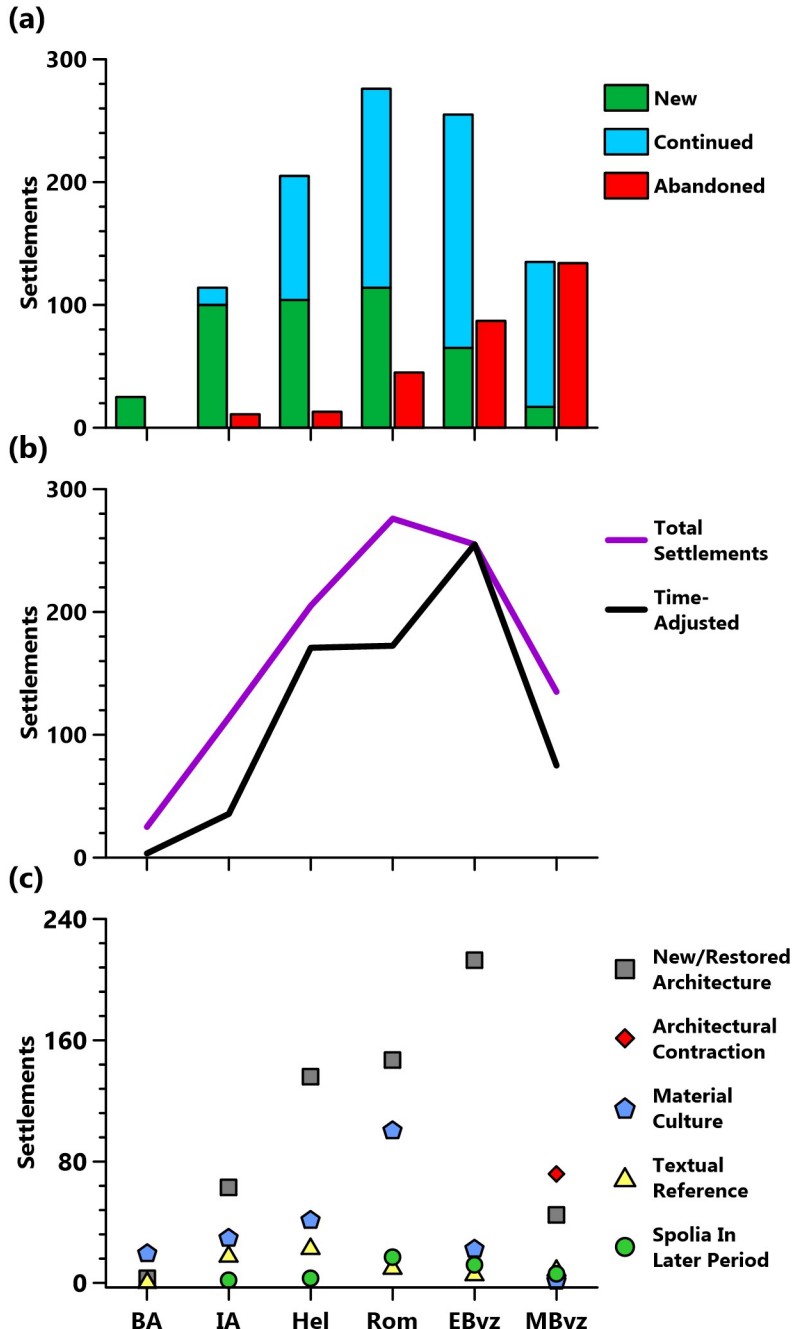

**Fig 2. Presentation of the TIB-derived dataset.** (**a**) Total settlement data, (**b**) time-adjusted data (**c**) settlement numbers by evidence type.

since 1800 CE, or dated by re-used building materials, easily recognizable within later buildings (e.g., Classical architectural sculpture or inscriptions recycled in Late Roman fortifications). The character of settlement evidence also varied by period (Figs 2 and 4).

Interpreting changes in the number and locations of settlements for each period presents significant challenges, associated with chronology, interpretive uncertainty, and preservation bias (see Data Critique). However, some patterns are still observable and broadly consistent

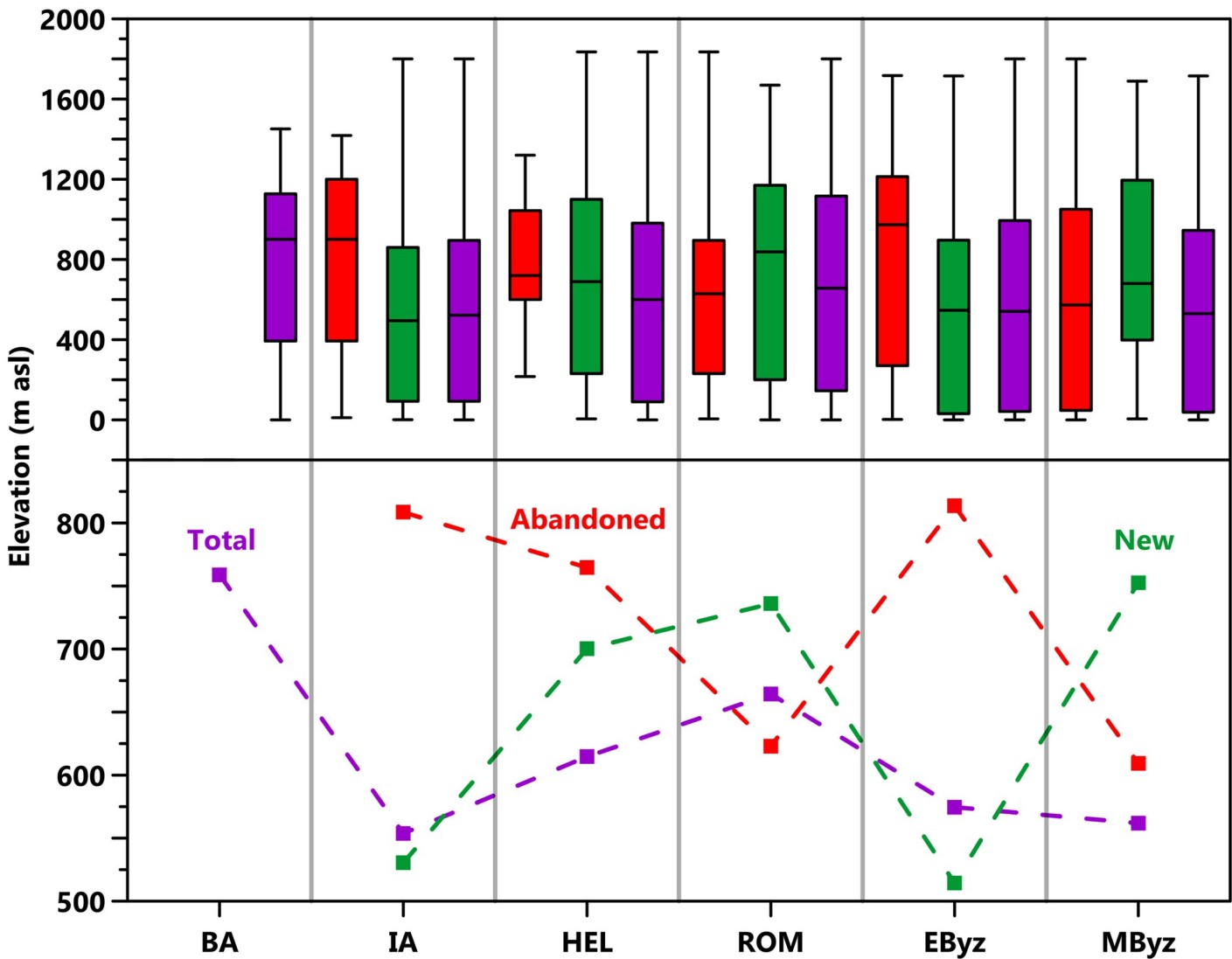

**Fig 3. Settlement elevation data by period.** Displayed as box-plots with inter-quartile ranges, median elevations (top) and mean elevations (bottom).

with the regional history, archaeological evidence, and paleoenvironmental data. The clearest of these trends is a steady increase and peak of settlement numbers in the Roman and Early Byzantine periods, followed by a significant reduction of Middle Byzantine evidence (Figs 2 and 3). These changes are consistent with data from across the Eastern Mediterranean [1, 57, 58] and are frequently hypothesized to result in part from changing climatic and environmental conditions. We discuss these hypotheses below, utilizing a regional case-study approach that includes comparison of our dataset with pre-existing evidence.

## Discussion

### History of climate and society in the Eastern Mediterranean

Studying associations between climatic and socio-cultural change, now termed the 'history of climate and society' (HCS; [59]) in the Eastern Mediterranean has been extensive, due to the region's historical significance, and a focus of many popular-science books [60–62]. Generally,

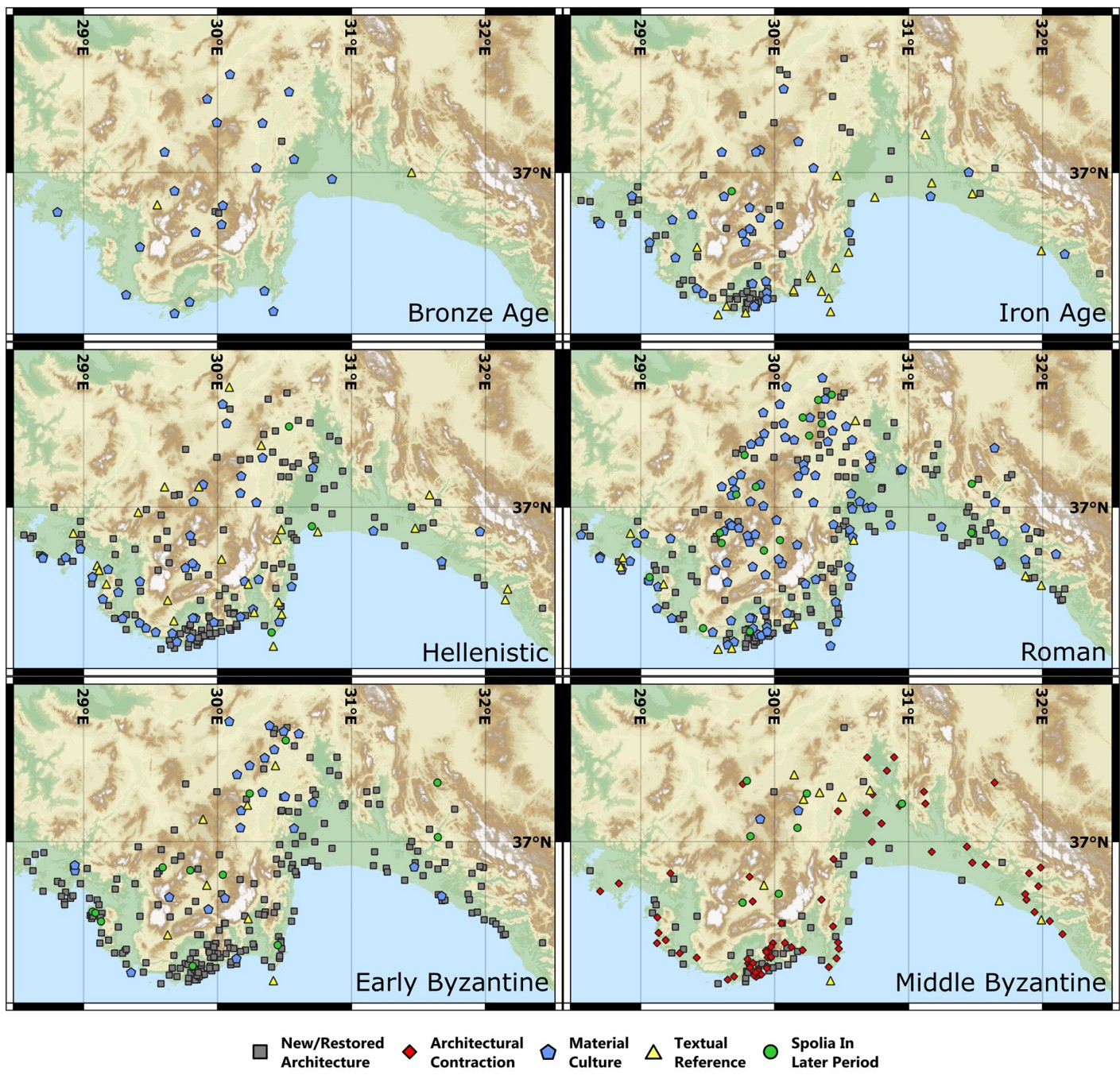

**Fig 4. Maps showing settlement evidence by period.** These were created in QGIS using the ASTER Global Digital Elevation Model v3 as a basemap [13].

such studies produce climate reconstructions which can be, linked to large-scale climatic periods or events (such as those displayed in Fig 5c), from a combination of paleoclimate proxy records, model simulations, and historical records. Assembled climate evidence is then compared to historical and archaeological evidence for social, economic, and cultural change, to assess human-climate-environment relationships. These studies intend to provide insight into past societal resilience and fragility, to prepare for potential impacts of future climate change

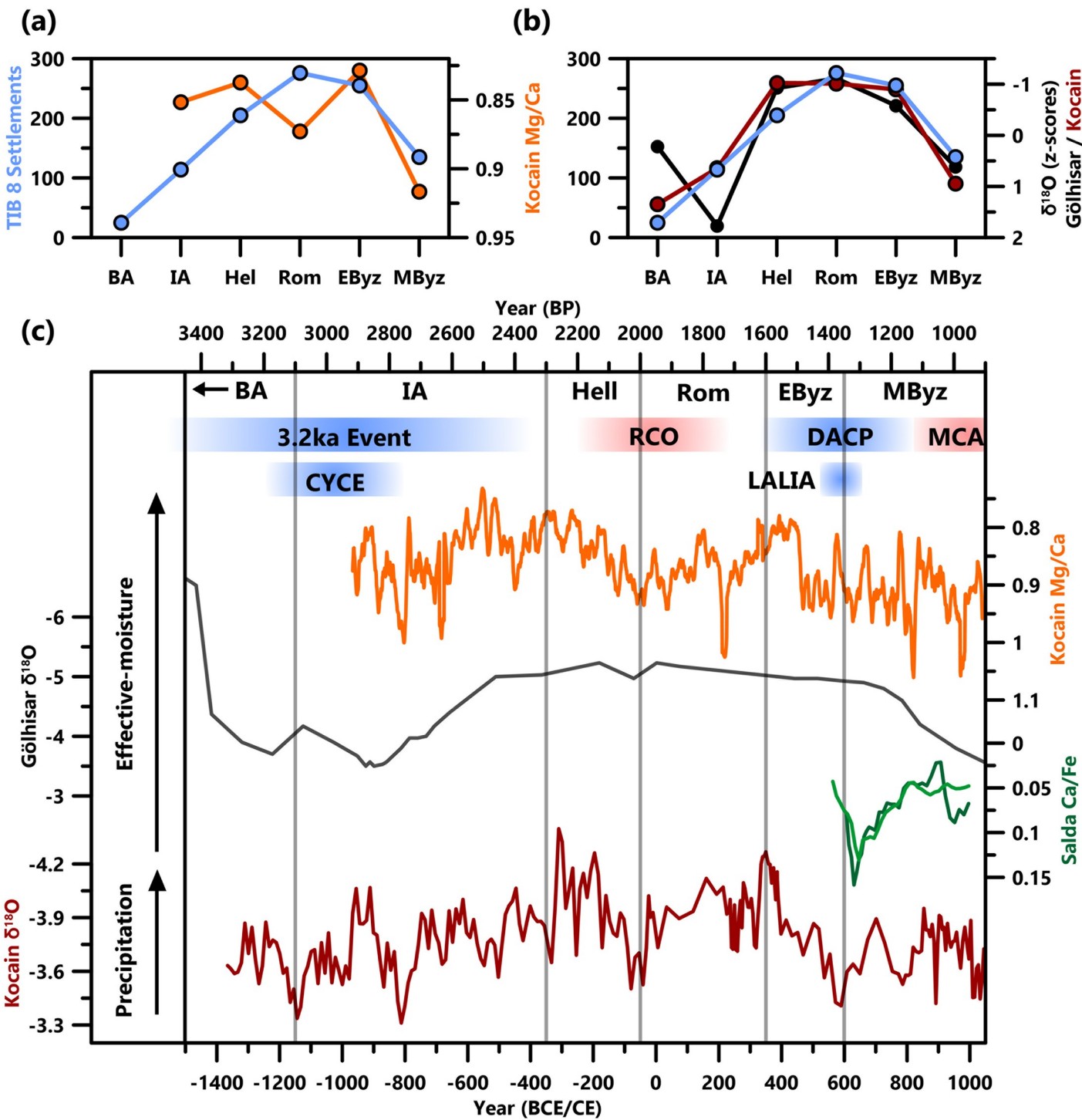

**Fig 5. Regional paleoclimate data.** Produced settlement data are compared to period averages of (**a**) Ko-1 Mg/Ca and (**b**) $\delta^{18}$O from Gölhisar and Ko-1. (**c**) Full paleoclimate data (metadata of records in S2 Table) with warm/cold intervals highlighted: Crisis Years Cold Event (CYCE), Roman Climatic Optimum (RCO), Dark Ages Cold Period (DACP), Late Antique Little Ice Age (LALIA) and Medieval Climate Anomaly (MCA). Kocain Mg/Ca (15-year averages), Gölhisar $\delta^{18}$O, and Salda Ca/Fe, are largely indicative of effective moisture; Kocain $\delta^{18}$O reflects precipitation, with secondary influences from seasonality and temperature [21–23].

[4, 63]. However, reviews of HCS studies [59, 64–67] have identified significant and recurring issues, namely (**1**) correlation-based conclusions that lack convincing causal explanations, (**2**) a bias towards periods of "crisis" which mischaracterizes human-environment interactions, (**3**) a focus on large regions without high-quality comparative datasets, leading to calls for "micro-regional" case-studies (stressed originally in [8]), and (**4**) the interdisciplinary challenge of comparing archaeological, paleo-environmental, and -climatic data of varied resolutions. These challenges are confronted in our presentation of regional data for climate and society in Lycia-Pamphylia.

HCS studies focused on the Eastern Mediterranean are numerous, with climate change frequently presented as an important factor driving prosperity and decline. In the sections below, we critique this over-simplified and generalized model, by examining three previous hypotheses. Firstly, the Roman Climatic Optimum (RCO; previously Roman Warm Period) is a proposed period of warm and wet conditions controversially referenced as a driving force behind the large-scale success of the Roman Empire [62]. Advantageous climatic conditions are suggested to have contributed to enhanced agricultural productivity, with associated economic, demographic, and settlement growth, between 200 BCE and 150 CE [62]. The 1st and 2nd centuries CE have long been suggested as a period of unprecedented prosperity (the "Pax Romana")–a concept recently critiqued and discussed in [68, 69]. Secondly, enhanced precipitation has been designated as an enabling factor for Central and Eastern Mediterranean prosperity in the 4th- early 6th centuries CE [1, 70–72]. During this period, increased interconnectivity between regions, higher agricultural productivity, population, and settlement numbers, alongside the expansion of rural settlement onto previously unoccupied marginal land, are suggested [54]. Thirdly, a period of cooling and/or drying, beginning in the 6th century CE, has been suggested as an important contributing factor for a decline across the Eastern Mediterranean and surrounding regions in the 6th and 7th centuries CE [73–75]. Epidemics and famines are theorized to have been caused by climatic deterioration, leading to demographic contraction and settlement abandonment through deaths and migration.

There are several key criticisms of these types of hypotheses, linked to the above broader criticisms of HCS studies. Most apparent is the over-simplified and large-scale nature of such hypotheses. Thus, care must be taken when discussing generalized patterns for broad regions or periods, as both climatic and socio-economic conditions have the potential for high variability on both spatial and temporal scales, especially in the Eastern Mediterranean [4, 8, 9, 12, 21]. Regional analyses are therefore required where the main climatic constraints on agricultural productivity are identified. For most regions in the Eastern Mediterranean, the basic assumption that higher temperatures and water availability will facilitate higher agricultural productivity is too simplistic [3, 19]. A regional-scale analysis of Lycia-Pamphylia is enabled by our new dataset of settlement character and location, and the existing high-quality paleo-environmental and -climatic records. Chronological and preservation biases in settlement datasets are limiting factors for this analysis (see Data Critique). However, the consistency of patterns across varied data-types, particularly for the Roman to Middle Byzantine periods, suggests they result from regional changes in settlement density and distribution. Comparing datasets of different temporal resolutions, such as highly-resolved paleoclimate records and period-averaged settlement data, presents significant challenges. Averaging of high-resolution paleoclimate data for archaeological periods or centuries is one methodology frequently utilized in the Eastern Mediterranean (e.g., [2, 58, 76]), here providing interesting results (Fig 5a and 5b). A visual relationship between $\delta^{18}O$ and settlement evidence is apparent (Fig 5b). However, averages can be misleading and conceal the high variability of climatic conditions within each period (Fig 5c). Additionally, agricultural productivity is more reliant on effective moisture, which for the higher-resolution Kocain Mg/Ca record [21] shows a weaker

relationship (note the Roman Period in Fig 5a). In the sections below, we consider the full picture of climatic variability during the periods under discussion and assess the extent of their impact on regional settlement location and character, in addition to the contribution of other factors.

## A Roman Climatic Optimum?

Roman-period (50 BCE– 350 CE) settlements in our dataset were largely continuous with the preceding Hellenistic period (350–50 BCE). A total of 45 Hellenistic settlements were abandoned and the remaining 160 constitute 58% of the 276 Roman period settlements. A shift in elevations is observed, with the highest mean elevation of all considered periods for new (736m asl) and total (664m asl) settlements (Fig 3). Abandoned settlements are more frequent at low elevations (37/45<1000m asl; mean = 622m asl) in southern and western Lycia, especially in association with older Iron Age and Hellenistic Lycian dynastic centers. For example, within 20km of Myra and Finike (ancient Phoinix), at least 10 settlements contain Hellenistic fortified tower-farmsteads or "turmgehöfte" [77] that show no signs of Roman occupation, though many were reoccupied in the drier Early Byzantine period (e.g., Beymelek, Davazlar, Gelemen or Pharroa). Of all new Roman settlements, a significant portion are located at mid-high elevations (68/116>600m asl), with highland concentrations in Pamphylia north of Side and Alanya and in northern Lycia/south Pisidia (Figs 3 and 4; S3 Fig). In the latter region, many rural Roman settlements are identified, primarily via inscriptions and ceramics, in surveys conducted by the Sagalassos Archaeological Research Project [78, 79]. Rural settlements constitute a large proportion of new Roman settlements (63 villages and farmsteads). Industrial rural estates based on the Italian villa rustica model appear at 10 settlements (e.g., at Çaylakkale, Gedelma, Hamaxia). Conversely, most Roman-period urban settlements (70/81) were continuations from the Hellenistic period.

Pollen data from around Sagalassos suggest that interior regions with high Roman-period settlement density in our dataset (and in the Sagalassos Hinterland survey [80]; Fig 6) experienced expansion of upland agriculture and arboriculture in the 1st century CE. At the high-elevation Bereket Basin site, OJCV values peak with significant increases also occurring at Ağlasun, Gölhisar, and Gravgaz [81]; Fig 6; S4 Fig). Despite proliferation of imperial-period architecture and material culture, and expansion of agricultural activity that matches with the RCO hypothesis, current evidence does not suggest that these were determined (or enabled) by climatic factors.

During the RCO, Mg/Ca ratios from Kocain Cave indicate that effective moisture was declining from ~100 BCE until ~30 BCE, then remained low until ~100 CE, with a further short-lived dry phase at ~220 CE also evidenced by a soot layer and pollen statistical analysis at Gravgaz Marsh [21, 82, 83]. Between these dry phases, Mg/Ca ratios were close to their mean values (0.88mmol/mol$^{-1}$) until ~120 CE and then lower (indicating slightly wetter conditions) until ~200 CE. $\delta^{18}$O values from both Kocain Cave and Lake Gölhisar suggest slightly wetter conditions from 50/1 BCE; however, this may result from the lower resolution of these archives which misses out shorter-term changes in climate. Slightly enhanced precipitation is suggested by $\delta^{18}$O from Kocain Cave during 150–200 CE, matching the wetter conditions suggested by Mg/Ca. This likely reflects conditions wetter than the average for the Roman period, but not as wet as the start of the preceding Hellenistic Period or the distinct period of high effective moisture in the 4th and 5th centuries CE (~330–460 CE; [21]). In our study region, during the RCO, and Roman Period, there was a high level of variability, with little evidence to suggest an enduring warm and wet phase. In fact, a lack of evidence for consistently warmer and/or wetter conditions during the RCO is observed throughout the Eastern Mediterranean

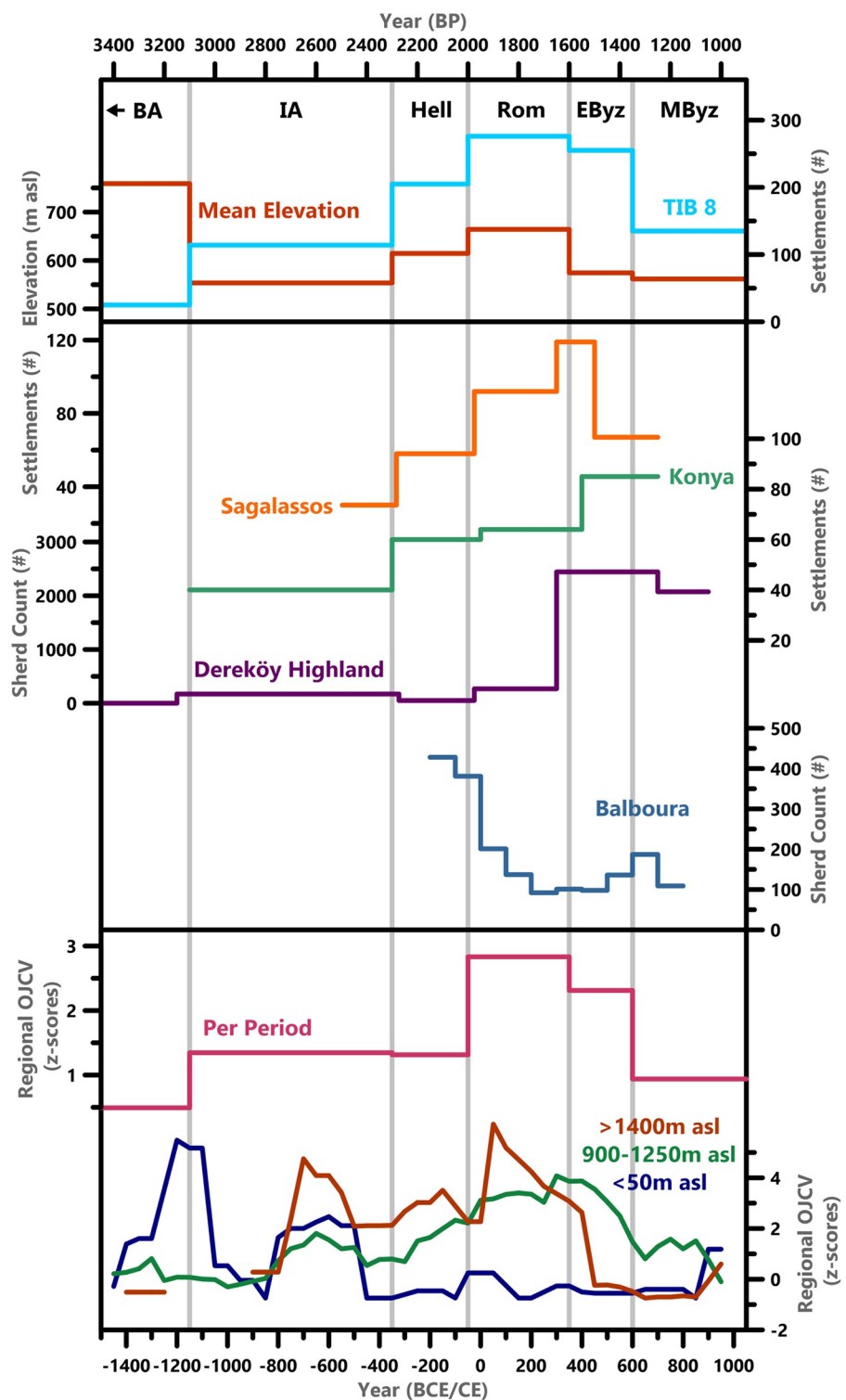

**Fig 6. Regional survey data.** TIB settlement data and mean elevation by period compared to other regional surveys and OJCV value z-scores—per period and as 200-year averages. References found in the main text.

[76, 84]. However, highly-resolved and well-dated paleo-temperature records are more sparsely distributed in the Eastern Mediterranean, and rarely extend back into the first millennium CE [4]. For example, the only paleo-temperature record in Lycia-Pamphylia starts in 1125 CE [85]. Further work is needed to reveal whether temperature changes influenced the previously described settlement shifts.

At least in our study region, hydroclimatic conditions were not the main factor responsible for the expansion of agricultural activity and settlement during the Roman period. Instead, we argue that integration of the region into the Roman imperial system after ~50 BCE was more important. A formal treaty of Lycia with Julius Caesar was signed in 46 BCE [86], under Augustus (27 BCE– 14 CE) the first colonies were established (e.g., at Kremna and Olbasa, and perhaps at Balboura), and final annexation occurred under Claudius in 43 CE [87, 88]. More than a century of heavy investment followed, which was financed both by the state and by local elites choosing to incorporate their lands into the Roman imperial system [89, 90]. Monumental agorai, theaters, baths, aqueducts, and temples, were rapidly constructed in both coastal and upland Roman settlements (Fig 7b; [87, 91]). Interconnectivity increased with infrastructural investment, and agricultural products critical for supply of Roman armies on the Danube and Syrian-Mesopotamian frontiers were more readily exported [92]. Roads were systematized, as evidenced by milestone inscriptions and the important Stadiasmus Patarensis inscription of regional road networks [93]. Lycian harbors became important nodes in regional grain markets and saw significant infrastructure upgrades. For example, granaries were built at Andriake and Patara, with a lighthouse at the latter [94]. Tax collection from cities was standardized across Roman territory and the state now profited from the rich resources of Lycia-Pamphylia, including traditional agricultural produce but also sponges, goat hair for ropes, wild animals for the circus, fish processed as garum, and, most importantly, timber [87, 88]. Agricultural efficiency for the Mediterranean triad (grapes, grains and olives) was enhanced with Roman innovation of the screw press for grapes and olives and the rotary mill for grain [95, 96]. Roman hegemony also brought newfound security from unrest and a regional threat of banditry and looters [88], and might be reflected by a reduction in both urban fortifications and tower-farmsteads, as noted earlier [97, 98]. All the above factors increased the importance of Lycia-Pamphylia for trade and enabled intensification of settlement and agriculture in the hinterlands [94, 99]. These developments occurred despite prevailing climatic conditions that likely made agriculture more challenging when compared to the preceding and succeeding periods. The dense settlement detected in the Roman period continues into the Early Byzantine period; however, a shift in the character and location of settlement is observed.

## Late Antique prosperity and adaptation?

Settlement numbers peaked in the Roman period (50 BCE-350 CE), with the majority (189/276 Roman settlements: 68%) continuing into the Early Byzantine period (350–600 CE). Overall settlement numbers remained high, with a slight decline in absolute numbers, with 255 of the total 381 settlements (67%) dated to the Early Byzantine period. When bias from period length is counteracted by a time-adjustment (see Data Critique), the Early Byzantine period experiences the settlement evidence peak (Fig 2). Important, however, is the shifting character and location of settlement between these two periods. There is a loss of settlement density in interior Lycia (Fig 4; S3 Fig) caused by the abandonment of Roman-period highland rural settlements, which contributed to the highest mean elevation of abandonments for any period (814m asl; Fig 3). Dense settlement on the coast continues, however, with 87 settlements <200m asl, including older Hellenistic and Roman cities (Fig 3; S3 Fig). New rural settlements also appear in the hinterlands of these cities, for instance around Fethiye, Ölüdeniz, and Myra;

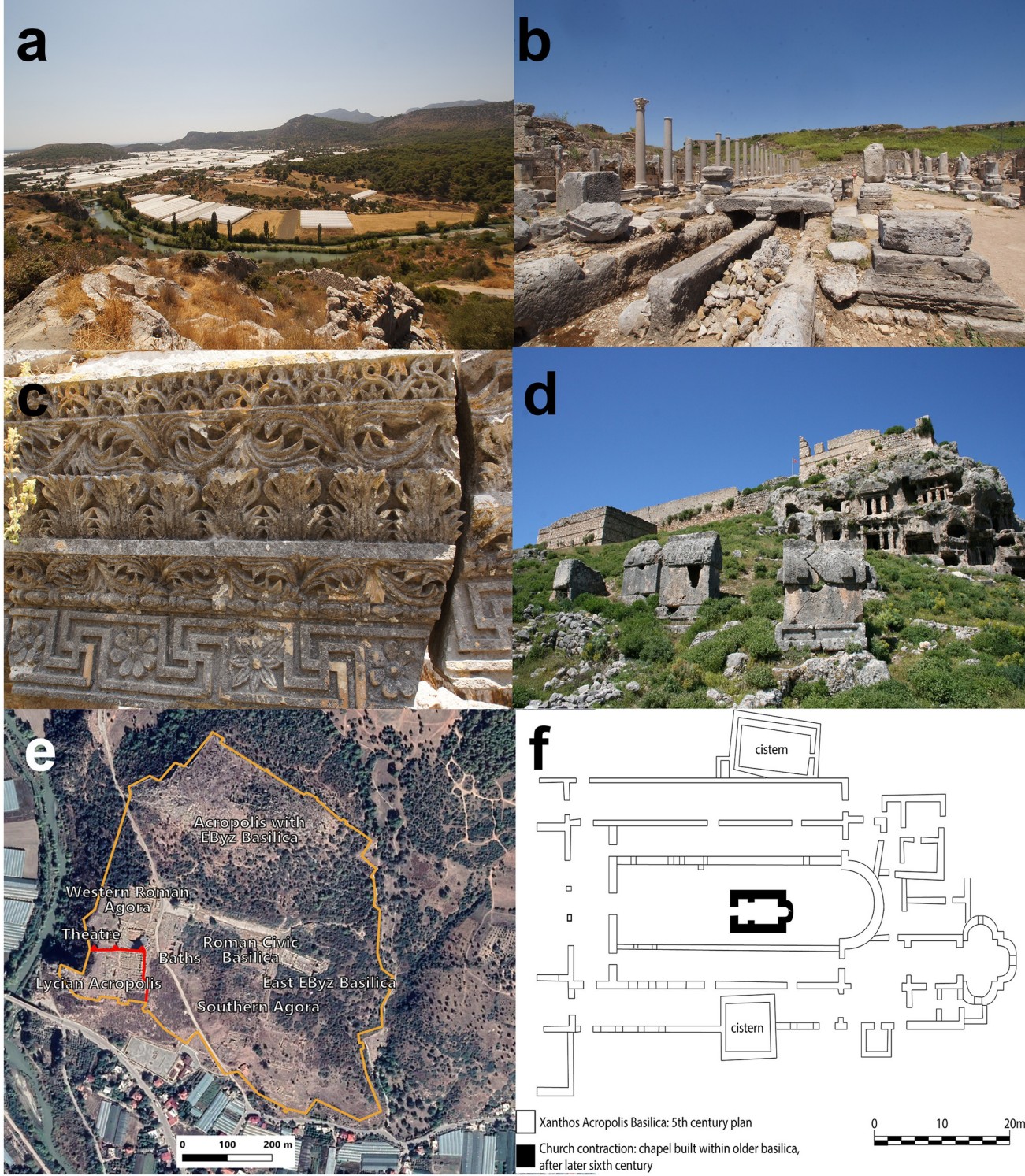

**Fig 7. Site photographs and plans from Lycia-Pamphylia.** All photographs and plans are by Jordan Pickett, excluding 7d. (**a**) A Lycian alluvial landscape, looking west over the river Eşen Çayı / anc. Xanthos from 7th-9th century Byzantine fortifications on the Lycian Acropolis at the city of Xanthos, with modern greenhouses below; (**b**) the Roman colonnaded street with cascading aqueduct-fed fountains at Perge; (**c**) Sixth century CE architectural sculpture from a church at Asarcık; (**d**) Tlos, with Roman stadium in the foreground, Iron Age-Hellenistic sarcophagi and tombs in the mid-ground, and Early-Middle Byzantine fortifications built atop the Iron Age acropolis (photograph by Hugh Elton); (**e**) Annotated satellite image of Xanthos, with Iron Age fortifications (in orange) and Middle Byzantine refortification of the Iron Age Lycian Acropolis (in red) added in QGIS; (**f**) plan of Middle Byzantine contraction of older church basilica on Xanthos Acropolis (after [37]).

the lowest mean elevation of new settlements for any period (514m asl) lowers the overall mean elevation (575m asl; Figs 3 and 4).

These findings are largely coherent with regional understanding. Though Martin Harrison [100] previously suggested that coastal cities in Lycia declined from the 5th century CE, with population and important settlements migrating inland to the mountainous regions, the inverse has otherwise been suggested—and is supported by our dataset—with increased settlement density in coastal regions and a possible decline in the interior regions [12, 36, 41, 101, 102]. The pollen data provides further support for this trend, with an Early Byzantine period decline in OJCV values at the high elevation Söğüt and Bereket Basin sites (Fig 6), coincident with the end of the BOP at these sites [17]. A decline in OJCV values is also observed in the other records from around Sagalassos (Ağlasun and Gravgaz Marsh), coincident with a reduction in settlement numbers in the hinterland survey (Fig 6; [80]). Similar to the Roman period, long-distance maritime transport of agricultural produce was important, with the state subsidizing supply of food staples both to Constantinople and to Byzantine armies campaigning against Sasanid Persia [103, 104]. This may have promoted expansion of settlements near to the coastlines and made inaccessible mountainside settlements comparatively less efficient [12, 71]. Because lower coastal regions have enhanced precipitation, and an extended growing season, compared to the higher elevation interior, they are more productive and allow for greater crop diversity [72]. The idea of a climate-driven period of "prosperity" in the 4th-6th centuries CE is therefore oversimplified, at least in Lycia-Pamphylia [12, 36].

Adam Izdebski et al. [1] examined Early Byzantine "prosperity" in relation to climatic conditions, concluding that enhanced precipitation contributed to the expansion of rural settlement and agriculture into environmentally marginal terrain. However, at that time the only available high-resolution paleoclimate records in Anatolia were those from Nar Gölü [105–107] and Sofular Cave [108], located in the Central Anatolian Plateau and Black Sea coastal regions, respectively. These records, and associated archaeological and historical data, suggest enhanced aridity in 350–470 CE, followed by wetter conditions [1]. However, the climate of Lycia-Pamphylia is now better understood due to the local Kocain Cave record [21], which stresses spatial heterogeneity of climate in Anatolia. The record demonstrates that the SW region experienced the inverse pattern to that previously suggested, sharing more similarities with the Aegean [21, 76]. Kocain $\delta^{18}$O and Mg/Ca ratios indicate very high precipitation and effective moisture between 330 and 400–460 CE, followed by a rapid shift to drier conditions that persisted until ~830 CE (Fig 5). This is important as high-resolution variability of climatic conditions in SW Anatolia were previously unknown for most of this period. The Lake Salda record, for instance, starts at ~560 CE, and there are no historical references to climatic conditions from the region [22, 23, 109].

Reduced effective moisture from the mid-5th century CE may also be reflected in regional archaeological evidence for urban water infrastructure adaptations. Construction and refurbishment of large vaulted reservoirs for water storage escalated from the mid-5th century CE, with large-scale projects at seven of the region's major cities, including the well-excavated sites of Sagalassos, Patara, Rhodiapolis, Side, and Xanthos [110–113]. At Sagalassos, a reduction in the supply output of the older Roman aqueducts is visible in changes made to the city's fountains during the 6th century, with outlets cut at lower basin levels in the Upper Agora, alongside construction of new metal and textile workshops. New public fountains were still constructed in Sagalassos; geochemical evidence from calcites in feeder channels indicates supply via rain-capture and snow-melt [110], contrary to traditional Roman patterns of urban water supply via spring-fed aqueducts [114, 115]. New settlements entirely reliant on rainwater capture also appear in the Early Byzantine period, for instance at Lyrboton Kome [10] and Gemiler Adası [116]. These urban water infrastructure adaptations coincide with the

demonumentalization and ruralization of cities, the appearance of church architecture, and renewal of fortifications described below. These economizing adaptations are traditionally associated in scholarship with "decline" but lately considered to be indicative of "adaptive reuse" [103], here possibly linked to drier climatic conditions revealed by the Kocain Cave multi-proxy record.

Early Byzantine settlement is most clearly evidenced by church architecture, which appears almost exclusively after the mid-5th century CE [10]. Intensified investment is indicated by the number of settlements evidenced by architecture (213/255 Early Byzantine settlements; Fig 2), comprised mostly of church construction and renewal of fortifications. Roman *poleis* were required to have bishops, necessitating construction of churches and episcopal complexes. Church construction was frequently coincident with the demonumentalization and ruralization of city settlements. Churches were often well-integrated into Roman street plans but incorporated building materials from recently demolished Roman structures. They signalled more than change in religious belief, indicating new political and economic realities in cities [55, 117]. Churches are also found in close proximity to agricultural developments in the countryside such as terracing (as at Asarcik, but unfortunately not dated; e.g., as in [118]) and in cities near artisanal installations (at Patara and Sagalassos), or large-scale water storage (at Xanthos and Gemiler Adası). At Xanthos, for instance, recent excavations in the western agora have revealed two churches, one at the agora's center and one built into its western portico, surrounded by workshops and water installations that facilitated artisanal activity, all dated to the 5th and 6th centuries CE (Fig 7e; [119]). Whilst new churches are not necessarily indications of communal wealth, they did require significant investment [71, 120]. This was especially true in Lycia where churches featured high-quality stone carving, even within village settings (e.g., north of Demre around Asarcık, Fig 7c; [121]). Lycian churches have long been a focus of scholarly attention (e.g., [48]), which, combined with their robustness, increases the archaeological visibility of rural settlements. These factors are perhaps reflected in the increase of rural settlements in our data between the Roman (43%: 119) and Early Byzantine (51%: 130) periods.

From the mid-5th century CE, older Roman cities saw growing industrialization throughout our region but began to lose their monumental aspect, alongside adaptation or reconstruction of older Roman public buildings (including theaters, baths, and agorai) for artisanal and industrial activities. For instance at Side, glass workshops appeared in the theater and around an abandoned temple in the agora after the 5th century [122, 123]. At Limyra, baths near the theater were abandoned in the later 5th century CE and became home to workshops for textiles, metal and/or glass, and bone-carving [124]. At Andriake, a workshop for the extraction of precious murex dye was built in the 6th century using spolia from an older church: massive piles of murex shells and more than 22,000 amphorae fragments of the 6th and 7th centuries, thought to be related to murex production and transport, remained there [125, 126].

Whilst reduced effective moisture appears in southwest Anatolia after ~450 CE, climatic shifts will not be homogenous across the Eastern Mediterranean. A complex web of interconnected natural and human factors for societal change in the Early and Middle Byzantine periods is suggested elsewhere to have its origin in the mid-6th century CE. These include cooling and drying associated with volcanic-induced reductions in solar irradiation, earthquakes, and epidemics, and insecurity resulting from conflicts between the Byzantine and Sasanian Empires, the Umayyad Caliphate, and nomadic groups. These are discussed next.

## Middle Byzantine abandonment and renucleation?

The transitional phase between our Early Byzantine (350–600 CE) and Middle Byzantine (600–1050 CE) periods, from c. 550 to 650 or 700 CE, shows a significant shift in the intensity

and character of settlement. Settlement density was reduced in the Middle Byzantine period, with a total of 135 settlements. Of the settlements with Early Byzantine-period evidence, 134/255 (52%) had been abandoned and only 17 new settlements are evidenced. An even more pronounced reduction in settlement density is suggested when the length of the longer period is considered, with time-adjustments (see Data Critique; Fig 2b).

Additionally, settlements were no longer occupied at the scales of the preceding Roman and Early Byzantine periods: in larger urban settlements elsewhere, a *citte ad isole* pattern is frequently observed, whereby small clusters of agro-industrial activity or "islands" are visible within the skeleton of older Roman and Early Byzantine cities, which were otherwise uninhabited. For example, Myra contained three separate fortified enclosures: one at the acropolis [127], one at the Church of St Nicholas, which was rebuilt as a domed basilica in the 8[th] century CE [128], and one at the city's harbor, Andriake [36]. Antalya, which became a center for the Byzantine navy and administration alongside significant investment in new architecture and fortifications may be one exception, though 20[th]-century development has prevented any archaeological conclusions [9]. Furthermore, over half (72/135) of the Middle Byzantine settlements are only evidenced by church contraction. Very small chapels, often $<3x5m^2$ and sometimes with datable frescoes, are erected inside older derelict structures, mostly much larger church complexes [37] (Fig 7f). Whilst church contractions could be indicative of the persistence of Christian holy spaces, they also suggest insecurity, inability, or unwillingness to invest, and demographic decline.

Investment in new construction or restoration is evidenced at only 45 settlements, a figure lower even than the Iron Age. This largely comprised smaller settlements where older Iron Age or Hellenistic circuits were refurbished (e.g., Idebessos, Tlos), and village settlements with small chapels constructed within older fortifications (e.g., at Asar Dağı, Trysa). Despite indications of both abandonment and reduced settlement intensity throughout our study region, continued investment in some of the larger cities suggests renucleation. Major church construction (e.g., at Antalya, Myra, Patara) and resettlement of disused urban acropolises (e.g., at Perge, Tlos, Xanthos) was accompanied by increased administrative importance [129]. Byzantine lead seals, which were used to secure documents and indicate information about the author [130], demonstrate the importance of Antalya, Myra, Perge, and Tlos as centers of Byzantine naval and fiscal administration in the Kibyrrhaeot theme [10, 131] (S3 Table).

Other regional surveys, relying primarily on ceramic evidence, show divergent patterns (Fig 6). In the Sagalassos Hinterland survey, settlement numbers had already declined for the period starting at ~450 CE, whereas in the Dereköy Highland sherd counts remained high in their period starting 700 CE [80, 132, 133]. In the Balboura survey, sherd counts increase in the 7[th] century CE, after re-dating of Cypriot Red Slip Ware (CRSW) by Armstrong [134]–a local/regional production that satisfies basic household needs [135]. In both surveys, the majority of Middle Byzantine sherds are local [136–139]. This pattern, which suggests a reduction in inter-regional exchange, is observed across the region, at settlements such as Sagalassos, Limyra and Xanthos [79, 103, 140]. This reduction in inter-regional exchange was accompanied by a reconfiguration in some contexts, away from Constantinople and towards the southeast: 7[th]-9[th] century CE deposits from Sagalassos contained LR7 amphora with fish remains from Egypt, and deposits from Limyra contained wares from Cyprus, Egypt, and the Levant [141].

The transition from the Early Byzantine to Middle Byzantine period also marks a major shift in the agricultural history of our study region and the Eastern Mediterranean more broadly. Intensive crop cultivation slowed considerably, pastoralism grew in importance, and re-wilding of the landscape occurred with pine forest recolonization [81, 142, 143]. At the pollen sites where the BOP had not already ended (Gölhisar, Ağlasun, Gravgaz), it did in the 7[th]-

8[th] CE (summarized in [17]). This is reflected in the OJCV values (Fig 6; S4 Fig), which are minimal during the Middle Byzantine period.

Climate may have some responsibility for these changes in historical society within our study region. The new Kocain Cave Mg/Ca record (Figs 5c and 8; [21]) suggests that in Lycia-Pamphylia, a rapid shift away from high effective moisture occurred ~460 CE and conditions remained relatively dry until at least 830 CE, except for short-lived breaks from aridity at ~571–578, 672–676 and 778–784 CE. Especially dry conditions occurred around the start of the 7[th] century CE, with a minimum of precipitation indicated by Kocain $\delta^{18}O$ at ~570–590 CE and high Salda Ca/Fe ratios between ~620 and 650 CE overlain on a 600–770 CE dry-phase (Fig 8) [23]. Very dry conditions are suggested again around the start of the 9[th] century CE, with Kocain Mg/Ca and $\delta^{18}O$ ratios consistently high between 790 and 830 CE and the start of a shift to less negative Gölhisar $\delta^{18}O$ ratios [21, 22]. The initial aridification precedes the 536 CE eruption that resulted in a 12–18-month dust-veil covering much of the northern hemisphere, chronicled by Byzantine, Syrian and Chinese sources [144–146]. In other regions, climatic deterioration is suggested to have followed cooling resulting from this eruption and two others (540 and 547 CE), reinforced by sea-ice feedbacks, leading to the Late Antique Little Ice Age ("LALIA": 536–660 CE). However, this is now thought to have lasted for a shorter period of time, "the 536–550 CE climate downturn" [73, 147]. The Kocain Cave record, and other Eastern Mediterranean records, perhaps support the suggestion that the 536 CE dust-veil event occurred superimposed over a longer-lasting pattern of cooling/drying climate known as the Dark Ages Cold Period ("DACP": 450–800 CE; [21, 148]). Records from elsewhere in Anatolia, in Lake Nar and Sofular Cave, show either wetter conditions or no indication of a climatic impact from the 536–550 CE downturn [105–108], further stressing regional and local climatic variability.

An extended dry period is suggested for Lycia-Pamphylia (~460–830 CE) that would have lowered potential agricultural productivity, potentially disenabling arboriculture at higher elevations, and may have contributed to socio-economic and settlement decline. However, it is important to consider that there were earlier instances of dry climatic conditions where the local population appears to have been relatively unafflicted, or even to have prospered. For example, during exceptionally dry conditions in the middle of the Iron Age (800 and 700 BCE), Greek colonization started in our region, and, during the RCO, agriculture thrived despite relatively low effective moisture. Additionally, the shift to drier conditions occurred at least a century earlier than the reduction in settlement numbers and investment, with a shift away from agriculture and towards pastoralism, which can be broadly dated to somewhere between 550 and 650 CE. This suggests that either the duration of the climatic change, or its combination with other factors was more important. These are primarily (1) pathogenic, (2) seismic, and (3) defensive (Fig 8).

(1).   The appearance of the Plague of Justinian after 541 CE and lasting into the 8[th] century CE, an early epidemic of the bubonic plague caused by the bacterium *Yersinia pestis*, was traditionally understood as a mass mortality event, especially during the 540s CE [62, 149]. Its presence in Lycia-Pamphylia is securely evidenced, with a detailed and contemporary account of widespread death at Myra, probably in 542 CE, that interrupted agricultural commerce from the Lycian interior to the port-city [109, 150], and aDNA of *Yersinia pestis* identified from human remains at Sagalassos [151]. However, the demographic impact of the pandemic has recently been questioned [152–154]. The historical account of Myra also indicates recovery of the rural landscape thereafter [109, 149, 150] and, with the exception of the single victim at Sagalassos, aDNA is lacking for now [151]. Late Antique plague may have been a greater stimulus for shifts in religio-cultural beliefs

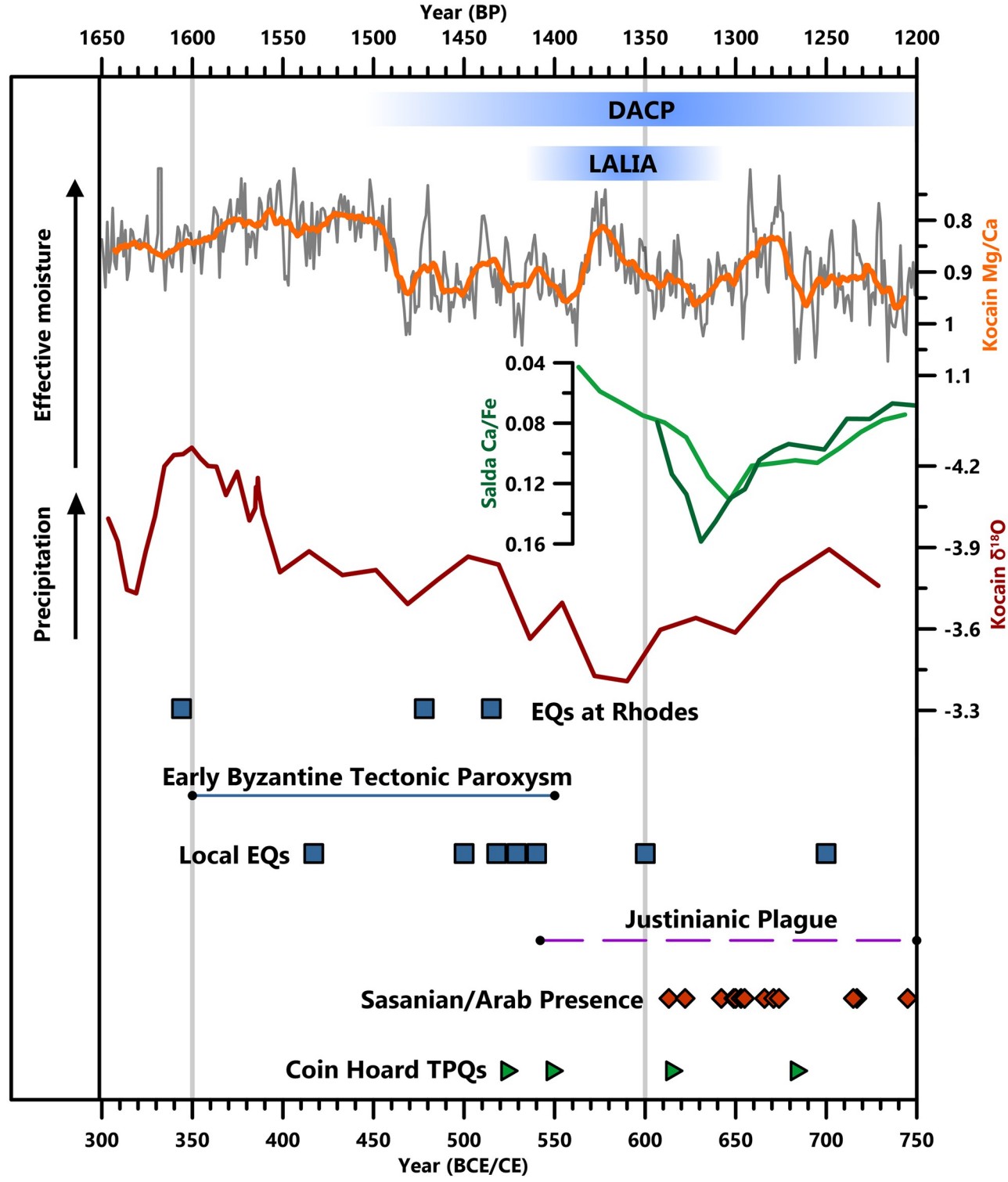

**Fig 8. Climate changes and other events during the Early Byzantine and Middle Byzantine periods.** The Ko-1 Mg/Ca record is displayed as annual (grey line) and 15-year (orange line) averages. Coin hoards, indicative of insecurity, are displayed by their *terminus post quems* (TPQs), indicating their earliest possible date of deposition (S3 Table).

and traditions than demographic or economic problems [155–157]. However, regional evidence indicates plague's presence, and it cannot be excluded as a factor in the widespread settlement abandonment, and possible demographic contraction, that is clearly visible in the archaeology of Lycia-Pamphylia between the mid-6th and 7th centuries CE.

(2).  Another major factor in 6th-century changes elsewhere, and evidenced in Lycia-Pamphylia, are earthquakes [75, 158] (Fig 8). The Aegean tectonic plate, a subduction zone, nearly parallel to and offshore from the south Anatolian coast [159], causes frequent earthquakes in Lycia-Pamphylia [160]. A clustering of major seismic activity around the Eastern Mediterranean between ~350 and 550 CE, known as the Early Byzantine Tectonic Paroxysm (EBTP; Fig 8), was caused by the reactivation of this subduction zone [161–163]. An early 6th-century (possibly 500 or 518 CE) earthquake occurred north of Antalya [164, 165], collapsing numerous light buildings and the monumental gates at Sagalassos [166, 167], with rebuilding activities widely visible in the city's archaeology [168]. In 528/9 CE, an earthquake centered on Fethiye-Meis [169] caused extensive damage across Lycia, particularly at Myra [170], with support given from imperial funds for reconstruction, according to typical protocols [158]. Communities suffering from earthquakes after the 6th century did not receive help or repair, however. Another earthquake around 600 CE "virtually wiped out Sagalassos as an urban settlement" [168] and further earthquakes in the 7th and 8th centuries have been observed in damaged and unrepaired structures at Side and around Myra [10].

(3).  Regional insecurity also increased from the 7th century. Towards the end of the Byzantine-Sasanian conflicts, the Sasanians captured nearby Tarsos in 613 CE and Rhodes from 622/623 CE. The Umayyad Caliphate's capture of Egypt in 642 CE ended the Annona grain supply to Constantinople, the transport of which relied on Lycian harbor cities. An Umayyad navy, formed in 649 CE, created coastal insecurity that endured for centuries. Byzantine naval hegemony ended with defeat at the Battle of the Masts in 655, and was followed by direct raids on our study region in 666, 671–673, 676–678, 715, 807 and 1034 CE [171] (Fig 8; S3 Table). As Douglas Baird [172] observes for the Konya Plain, investment in land for agriculture would quickly have become unstable upon the mere threat of seasonal Umayyad raids, which could have disrupted the economy and population without the need for destruction of property. An interesting suggestion, beyond the scope of this paper, is that the complex web of interconnected factors (including climatic deterioration), which impacted varied regions across the northern hemisphere, may have "pushed" or "enabled" invasions by the Rashidun/Umayyad Caliphates [75], and nomadic groups [73].

Overall, evidence from Lycia-Pamphylia indicates settlement abandonment and contraction suggestive of a population reduction, a shift from BOP agriculture to pastoralism, and the presence of dry climatic conditions, plague, earthquakes, and insecurity—all present at the transition between our Early and Middle Byzantine periods ("end of Late Antiquity") after the 6th century CE. However, clear causal attribution between archaeologically-visible developments and the aforementioned factors remain difficult to determine throughout Anatolia [173]. Evidence is overall stronger for the natural hazards, when compared to the evidence for societal response to them. Better chronological precision for archaeological change and site-specific studies of proxies for societal response (e.g., urban trash mounds [6]) are required to disentangle the causes of this change. Additionally, as with climate, there are earlier instances of plagues, earthquakes, and conflicts, where the population of Lycia-Pamphylia appear to have been unafflicted (see [158, 174] for earthquakes). This emphasizes these factors were

contributory, rather than ultimate, and that it is fundamentally socio-political factors that determine the region's resilience and ability to adapt.

## Conclusions

The regional record of historical settlements provided by the TIB 8 provides a clear picture of settlement change for a region with previously disjointed evidence. Based on comparison of this dataset with proximate climatic and environmental proxies, we argue that in our study region climate change does impact potential agricultural productivity, which can lead to changes in settlement density and locations. However, the relative impact is highly variable, and socio-political factors are often more important. Settlement numbers peak between the 1st century BCE and the 6th century CE in the Roman and Early Byzantine periods, alongside evolutions in the character and locations of settlement that begin ~460 CE, followed by a clear decline in absolute numbers that continues into the Middle Byzantine period. These shifts occur prior to the LALIA, increased seismic activity and the beginning of the Justinianic Plague, though they appear to have been intensified by the latter.

We find that, during the "Pax Romana" (1st and 2nd centuries CE), increasing agricultural productivity, settlement numbers, and population, with economic investment, does occur. However, linking these developments to warmer and wetter climatic conditions (as in the Roman Climatic Optimum [62]) is not possible in our study region, where conditions were drier than in the preceding and succeeding periods. During a previously proposed period of enhanced winter precipitation in the 4th-6th centuries CE [1], Lycia-Pamphylia experienced both enhanced effective moisture between 330 and 460 CE and dry conditions from 460 CE onward. This shift was concurrent with a change in the character of settlement, with urban water infrastructure possibly changing as an adaptation to drier conditions, demonumentalization and ruralization of cities that became agro-industrial villages, the appearance of church architecture, and renewal of fortifications. Drier conditions continued until at least 830 CE, and other factors with their origins in the mid-6th century CE, including recurrent plague epidemics, earthquakes and interstate conflict contributed to a significant decline in settlement numbers. Altogether, the individual contribution of each factor is hard to quantify and the combination of many factors into a "perfect storm" appears to be more important. The greatly reduced number of surviving Middle Byzantine settlements exhibit contraction of church complexes, occupied areas, and investment, suggestive of population decline or changing societal priorities, though renucleation also occurs in some large centers.

Overall, we demonstrate that simple correlations between favorable (wetter) or adverse (drier) climatic conditions with positive or negative socio-economic conditions have numerous caveats, especially considering the spatial heterogeneity of both climatic and societal change in the Eastern Mediterranean. Lycia-Pamphylia flourished during the drier Roman period, evolved during another period of aridity after 460 CE, but suffered under the weight of multiple pressures (political, environmental-climatic, seismic, pathogenic) after the mid-6th century CE.

Examining human-climate-environment interactions has been demonstrated as requiring strong archaeological and historical evidence, as well as high-quality paleo-environmental and -climatic datasets. Further work can be done for earlier periods in this archaeological dataset. For example, interesting settlement, environmental, and climatic changes are identified in the Iron Age and Hellenistic period. Additionally, similar archaeological datasets can be produced in existing and upcoming TIB regions [175], many of which contain highly-resolved speleothem and lake paleoclimate records suitable for similar analysis: Cappadocia (Nar Lake [105]), Nicopolis and Cephalonia (Lake Trichonida [176]), Thrace (Uzuntarla Cave [177]),

Bithynia (Sofular Cave [108, 178]) and the Northern Aegean (Skala Marion Cave [179]). Furthermore, the Kocain Cave Mg/Ca record (and other high-resolution effective moisture records) may enable agroecosystem modelling of agricultural productivity for these regions (as in [39, 180]).

## Supporting information

**S1 Fig. Resolution pie charts.**
(TIF)

**S2 Fig. Number of periods evidenced.**
(TIF)

**S3 Fig. Settlement change heatmaps.**
(TIF)

**S4 Fig. Full OJCV data.**
(TIF)

**S1 Table. Settlement dataset.**
(XLSX)

**S2 Table. Paleo-data.**
(XLSX)

**S3 Table. Additional Byzantine-period evidence.**
(XLSX)

**S1 File. Inclusivity in global research.**
(DOCX)

## Acknowledgments

This paper was only possible due to the hard work of the Austrian Academy of Sciences and members of the *Tabula Imperii Byzantini* (https://tib.oeaw.ac.at/) team, the authors would like to thank them for their excellent dataset upon which ours was based—particularly the authors of volume 8, Friedrich Hild and Hansgerd Hellenkemper. Assistance with EPD (http://www.europeanpollendatabase.net) pollen data provided by Jessie Woodbridge, Henk Woldring and Nils Broothaerts is greatly appreciated. We would also like to thank John Haldon and Michael E. Smith for reading earlier versions of this manuscript. We appreciate the constructive feedback provided by the two reviewers and editor assigned to this article.

## Author Contributions

**Conceptualization:** Matthew J. Jacobson, Jordan Pickett.

**Data curation:** Matthew J. Jacobson, Jordan Pickett.

**Formal analysis:** Matthew J. Jacobson, Jordan Pickett.

**Funding acquisition:** Matthew J. Jacobson, Jordan Pickett.

**Investigation:** Matthew J. Jacobson, Jordan Pickett.

**Methodology:** Matthew J. Jacobson, Jordan Pickett.

**Project administration:** Matthew J. Jacobson.

**Resources:** Matthew J. Jacobson, Jordan Pickett.

**Supervision:** Alison L. Gascoigne, Dominik Fleitmann, Hugh Elton.

**Visualization:** Matthew J. Jacobson.

**Writing – original draft:** Matthew J. Jacobson, Jordan Pickett.

**Writing – review & editing:** Matthew J. Jacobson, Jordan Pickett, Alison L. Gascoigne, Dominik Fleitmann, Hugh Elton.

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
