## [Decision Letter · Decision Letter 0]

7 Mar 2022

PONE-D-22-00644Settlement, environment, and climate change in SW Anatolia: dynamics of regional variation and the end of AntiquityPLOS ONE

Dear Dr. Jacobson,

Thank you for submitting your manuscript to PLOS ONE. After careful consideration, we feel that it has merit but does not fully meet PLOS ONE’s publication criteria as it currently stands. Therefore, we invite you to submit a revised version of the manuscript that addresses the points raised during the review process.

the two reviewers appreciated your approach, results and interpretation, but one of them suggest several major to moderate revisions to improve the general clarity of your manuscript. I agree with this comments and I suggest you to consider them when revising your manuscript. Especially, take care of comments on the interpretation of climatic data.

We look forward to receiving your revised manuscript.

Kind regards,

Andrea Zerboni, Ph.D.

Academic Editor

PLOS ONE

Journal Requirements:

3. Please include a complete copy of PLOS’ questionnaire on inclusivity in global research in your revised manuscript. Our policy for research in this area aims to improve transparency in the reporting of research performed outside of researchers’ own country or community. The policy applies to researchers who have travelled to a different country to conduct research, research with Indigenous populations or their lands, and research on cultural artefacts. The questionnaire can also be requested at the journal’s discretion for any other submissions, even if these conditions are not met.  Please find more information on the policy and a link to download a blank copy of the questionnaire here: https://journals.plos.org/plosone/s/best-practices-in-research-reporting. Please upload a completed version of your questionnaire as Supporting Information when you resubmit your manuscript.

(This work was supported by the AHRC South, West and Wales Doctoral Training Partnership (Grant AH/L503939/1 to M. J. Jacobson). The funders had no role in study design, data collection and analysis, decision to publish, or preparation of the manuscript.)

6. We note that Figures 1 and 3 in your submission contain map images which may be copyrighted. All PLOS content is published under the Creative Commons Attribution License (CC BY 4.0), which means that the manuscript, images, and Supporting Information files will be freely available online, and any third party is permitted to access, download, copy, distribute, and use these materials in any way, even commercially, with proper attribution. For these reasons, we cannot publish previously copyrighted maps or satellite images created using proprietary data, such as Google software (Google Maps, Street View, and Earth). For more information, see our copyright guidelines: http://journals.plos.org/plosone/s/licenses-and-copyright.

a. You may seek permission from the original copyright holder of Figures 1 and 3 to publish the content specifically under the CC BY 4.0 license.  

Reviewers' comments:

Reviewer's Responses to Questions

**Comments to the Author**

1. Is the manuscript technically sound, and do the data support the conclusions?

Reviewer #1: Yes

Reviewer #2: Yes

2. Has the statistical analysis been performed appropriately and rigorously? 

Reviewer #1: N/A

Reviewer #2: Yes

3. Have the authors made all data underlying the findings in their manuscript fully available?

Reviewer #1: Yes

Reviewer #2: Yes

4. Is the manuscript presented in an intelligible fashion and written in standard English?

Reviewer #1: Yes

Reviewer #2: Yes

5. Review Comments to the Author

Reviewer #1: The paper “Settlement, environment, and climate change in SW Anatolia: dynamics of regional variation and the end of Antiquity” by Matthew et al. has the merit of making a large synthesis from different sources of data in a large area. The topic is consistent with the aims of PLOS ONE.

The paper, illustrating a possible link between climatic variations and change in settlements distribution in south-western Anatolia, is well written and provide a wide archaeological dataset. The analyses are well done.

In this kind of work, it is necessary to make some simplifications / assumptions that the authors rightly point out in the paragraph "Data critique".

I really appreciated this paper and I think that it is capable of improving knowledge on a key topic for this reason I think it is worth of publication in PLOS ONE after moderate revisions.

Main concerns:

I suggest to change a little bit the introduction (see the pdf). Specifically, the last part of introduction seems more a conclusion than the presentation of the focus of the paper.

In methods section, Authors should indicate the typology of satellite images and software GIS used.

In the last part of the discussion the Authors introduce non-climatic reasons for the change in the settlements distribution.

In my opinion these data need more references throughout the text.

Moreover, being a crucial and delicate point, I suggest improving the discussion on this aspect. In my opinion, in order to make clearly visible possible correlations, it could be very useful a dedicate figure on main non-climatic factors (e.g. socio-political reasons, seismicity, pandemics episodes, etc.) or the insertion of these data in figure 6 with climatic data, settlements and chronology. I really think this part could improve the paper, that I find interesting.

Figure 6 is one of the most important of the paper, but I don't understand the split in three parts and, in any case, in order to improve the readability, I suggest to use the same scale for x axis in the three figures.

In addition, I think that the paper might be a little shortened.

Minor concerns:

It seems to me that there are some mistakes in the ref to number of tables in the text. Please explain better table1 and add the years corresponding to each period in table 2 in order to make the paper more readable for non-archaeologists (I recognize that they are reported in the text, but I think could improve the understand of the paper put the years also in table).

The caption of figure 2 have to be improved (also including some description currently reported in caption of figure 3 –e.g. the meaning of typology of settlements).

About Figures:

Figure 2 please, improve the caption adding the explanation of terminology used in 2a and also explaining better the figure 2b

Figure 3 Is a good figure but due to the scale the symbols are not readable. I suggest changing the colour of “New restored architecture” too similar to “Material culture” and the colour of “Architectural Contraction” too similar to “Textual reference”.

Figure 4e I suggest increasing the thickness of the orange line

Figure 6 It is one of the most important of the paper, please see what I suggest above.

Reviewer #2: The present paper discusses the (often complex) correlation between environmental fluctuations and change in settlement structures. It focuses primarily on Southern Anatolia during the Roman and post-Roman periods. The article builds upon a very strong methodology and it makes use of a robust and solid set of data. I agree with the author(s) over the difficulty in exclusively related phenomena of contraction/expansion of settlements and climatic variations. Given the multi-faceted aspect of several paleo-climatic proxies, imputing solely to the natural factor dynamics of urbanization, ruralization, and land exploitation might be too much deterministic, in my view. And yet, the approach taken by the authors does not simply dismiss the environmental cause, but rather they frame it alongside with societal, political, and (very much appreciated) technological transformations. All these causes contribute to read the Southern Anatolian landscape transformation in a fuller and more comprehensible way.

6. PLOS authors have the option to publish the peer review history of their article (what does this mean?). If published, this will include your full peer review and any attached files.

Reviewer #1: **Yes: **Monica Bini

Reviewer #2: **Yes: **Rocco Palermo

---

## [Author Response · Author response to Decision Letter 0]

18 Apr 2022

Journal Requirements –

1. PLOS ONE Style requirements

Manuscript has been re-formatted according to PLOS One guidelines

2. Permits for work

No permits required, statement added to end of methodology: “No permits were required for the described study, which was exclusively desk-based and complied with all relevant regulations.”

3. Inclusivity questionnaire

Included in re-upload

4. Amended funding statement

Included in cover letter

5. Data availability statement

Included in cover letter

6. Possibly copyrighted map images

Maps have now been updated to be based on the ASTER Global DEM v3 which is cited in figure captions.

Reviewer #1 –

Changes to introduction – moved the end of the introduction to the conclusion 

Indicate typology of satellite images and software GIS used – satellite images from Google Earth used for identification, but not included in the figures. QGIS software now mentioned in methods section and in relevant figure captions.

Non-climatic reasons referenced throughout – relevant non-climatic reasons are referenced throughout the text, the factors mentioned at the end of the discussion are only relevant for the final section. 

Non-climatic factors figure – New figure for the E-Byz to M-Byz transition added (Fig 8)

Paleoclimate figure split into 3 parts – this figure is separated to emphases a few points in the first part of the discussion about comparing paleoclimate to archaeological data. (a) and (b) compare the period-averaged climate data (which we say can be misleading) to the settlement data (which is lower resolution). (a) and (b) are separated to show that the correlation between precipitation proxies and settlements can be misleading, considering that effective moisture (evidenced in Mg/Ca ratios) is a better predictor of agricultural productivity. No changes were made to the figure, the reason for the split is explained in the paragraph preceding “A Roman Climatic Optimum?”

Shortening the paper – we don’t think anything can be removed from the paper without weakening the arguments made. The “Data Critique” section could be removed, or added to the supplementary; however, reviewer #1 likes this section and we agree it should remain included in the body of the text. Furthermore, this kind of honest critique section is normally confined to the supplementary so that it isn’t seen – we want to be as transparent as possible with our dataset.

Mistakes to table and figure numbers – corrected. 

Caption of figure 2 to describe “New” etc. – This is now included in the caption for Table 1 (settlement metadata), which comes long before Figure 2 in the text. This statement could also be added to the figure caption if needed: 

“New” settlements are those with evidence in the period, but no evidence in the preceding period; “Continued” settlements are those with evidence in the period and preceding period; “Abandoned” settlements are those with no evidence in the period, but evidence in the preceding period.”

Figure 3 (now Fig 4) scale not readable – corrected by changing colour and shape of symbols, also made the figure fit better on an a4 page so that the individual maps are larger. 

Figure 4e (now figure 7e due to re-ordering) line thickness – red and orange lines now as thick as they can be without obscuring the image. Writing is also larger and scale-bar more obvious with white background.

---

## [Decision Letter · Decision Letter 1]

8 Jun 2022

Settlement, environment, and climate change in southwest Anatolia: Dynamics of regional variation and the end of Antiquity

PONE-D-22-00644R1

Dear Dr. Jacobson,

We’re pleased to inform you that your manuscript has been judged scientifically suitable for publication and will be formally accepted for publication once it meets all outstanding technical requirements.

Kind regards,

Andrea Zerboni, Ph.D.

Academic Editor

PLOS ONE

Additional Editor Comments (optional):

The reviewers appreciated your manuscript and they suggest for immediate acceptance. I agree with them and I think your piece will be very important for regional studies and may represent a good example of geoarchaeological investigation on settlement distribution based on a multi proxy approach.

Reviewers' comments:

Reviewer's Responses to Questions

**Comments to the Author**

1. If the authors have adequately addressed your comments raised in a previous round of review and you feel that this manuscript is now acceptable for publication, you may indicate that here to bypass the “Comments to the Author” section, enter your conflict of interest statement in the “Confidential to Editor” section, and submit your "Accept" recommendation.

Reviewer #1: All comments have been addressed

2. Is the manuscript technically sound, and do the data support the conclusions?

Reviewer #1: Yes

3. Has the statistical analysis been performed appropriately and rigorously? 

Reviewer #1: N/A

4. Have the authors made all data underlying the findings in their manuscript fully available?

Reviewer #1: Yes

5. Is the manuscript presented in an intelligible fashion and written in standard English?

Reviewer #1: Yes

6. Review Comments to the Author

Reviewer #1: Most of the required changes have been correctly addressed and, in my opinion, the manuscript has been adequately improved.

I think that the manuscript could be published in PLOS ONE

7. PLOS authors have the option to publish the peer review history of their article (what does this mean?). If published, this will include your full peer review and any attached files.

Reviewer #1: **Yes: **Monica Bini

---

## [Editor Report · Acceptance letter]

17 Jun 2022

PONE-D-22-00644R1 

Settlement, environment, and climate change in SW Anatolia: Dynamics of regional variation and the end of Antiquity 

Dear Dr. Jacobson:

I'm pleased to inform you that your manuscript has been deemed suitable for publication in PLOS ONE. Congratulations! Your manuscript is now with our production department. 

Kind regards, 

on behalf of

Prof. Andrea Zerboni 

Academic Editor

PLOS ONE